# The Role of Imaging in Cervical Cancer Staging: ESGO/ESTRO/ESP Guidelines (Update 2023)

**DOI:** 10.3390/cancers16040775

**Published:** 2024-02-14

**Authors:** Daniela Fischerova, Filip Frühauf, Andrea Burgetova, Ingfrid S. Haldorsen, Elena Gatti, David Cibula

**Affiliations:** 1Gynecologic Oncology Centre, Department of Gynaecology, Obstetrics and Neonatology, First Faculty of Medicine, Charles University and General University Hospital in Prague, 121 08 Prague, Czech Republic; filip.fruheauf@vfn.cz (F.F.); david.cibula@vfn.cz (D.C.); 2Department of Radiology, First Faculty of Medicine, Charles University and General University Hospital in Prague, 121 08 Prague, Czech Republic; andrea.burgetova@vfn.cz; 3Mohn Medical Imaging and Visualization Centre (MMIV), Department of Radiology, Haukeland University Hospital, N-5021 Bergen, Norway; ingfrid.haldorsen@uib.no; 4Section for Radiology, Department of Clinical Medicine, University of Bergen, 5020 Bergen, Norway; 5Department of Biomedical Science for Health, University of Milan, 20133 Milan, Italy; elena.gatti2@unimi.it

**Keywords:** cervical cancer, staging, ultrasound, MRI, CT, PET-CT, neoplasm, diagnostic imaging

## Abstract

**Simple Summary:**

Constant technological development of modern imaging has led to substantial improvement in management and decision-making in the diagnostic and prognostic process of many different neoplasms. This also applies to cervical cancer. The main evidence, providing the base of recently updated ESGO-ESTRO-ESP recommendations (2023) on the management and treatment of cervical cancer, has been evaluated and reviewed in this paper. Ultrasound has been suggested as a valid alternative to MRI in primary diagnostic workup of cervical cancer if performed by an expert sonographer. Additionally, CT or PET/CT exhibits a substantial role in assessing the extrapelvic spread of the disease in locally advanced cases or when suspicious lymph nodes are detected. The purpose of this article is to provide a comprehensive review of the role of different imaging techniques in staging settings, displaying a focused interest in the use of ultrasound.

**Abstract:**

Following the European Society of Gynaecological Oncology (ESGO), the European Society for Radiotherapy and Oncology (ESTRO), and the European Society of Pathology (ESP) joint guidelines (2018) for the management of patients with cervical cancer, treatment decisions should be guided by modern imaging techniques. After five years (2023), an update of the ESGO-ESTRO-ESP recommendations was performed, further confirming this statement. Transvaginal/transrectal ultrasound (TRS/TVS) or pelvic magnetic resonance (MRI) enables tumor delineation and precise assessment of its local extent, including the evaluation of the depth of infiltration in the bladder- or rectal wall. Additionally, both techniques have very high specificity to confirm the presence of metastatic pelvic lymph nodes but fail to exclude them due to insufficient sensitivity to detect small-volume metastases, as in any other currently available imaging modality. In early-stage disease (T1a to T2a1, except T1b3) with negative lymph nodes on TVS/TRS or MRI, surgicopathological staging should be performed. In all other situations, contrast-enhanced computed tomography (CECT) or 18F-fluorodeoxyglucose positron emission tomography combined with CT (PET-CT) is recommended to assess extrapelvic spread. This paper aims to review the evidence supporting the implementation of diagnostic imaging with a focus on ultrasound at primary diagnostic workup of cervical cancer.

## 1. Introduction

Until recently, the clinical staging relied on clinical evaluation, preferably by an experienced examiner [1]. Conventional procedures, such as cystoscopy, proctoscopy, intravenous urography, and X-ray of the lungs, were performed [1]. Examination under anaesthesia was recommended, especially when the examination was difficult, or there was uncertainty regarding the involvement of the vagina, parametrium, or pelvic wall [2]. However, physical examination alone is known to have low accuracy for assessing tumour size and parametrial infiltration. Historically, Innocenti et al. reported sensitivities of clinical evaluation and transrectal ultrasound for diagnosing parametrial involvement, using surgico-pathology as the reference standard, of 52% and 78%, respectively [3]. Twenty years ago, the multicentric clinical trial of the American College of Radiology Imaging Network/Gynecologic Oncology Group (ACRIN/GOG183) enrolled 208 patients (from 25 centres, with invasive cervical cancer proved by biopsy results) who underwent pelvic magnetic resonance imaging (MRI) and contrast-enhanced computed tomography (CT) before definitive radical hysterectomy [4]. Correlation between maximum histopathologic tumour size and tumour size from clinical assessment (rs = 0.37, low correlation), CT (rs = 0.45, moderate correlation), and MRI (rs = 0.54, moderate correlation) were reported with the highest figures for MRI [4]. It is obvious that clinical assessment alone is insufficient to accurately assess the size of smaller tumours that do not cause cervical enlargement. In the same ACRIN/GOG 183 study, MRI yielded higher sensitivity (53%) than clinical assessment (29%) for diagnosing parametrial invasion [5]. A systematic review including 3254 patients also found that MRI had much higher sensitivity than clinical examination for diagnosing parametrial invasion (sensitivity: 84% vs. 40%) and locally advanced disease (79 vs. 53%) [6]. Clinical understaging can be caused by the inability to detect incipient parametrial invasion by clinical examination, especially ventrally and/or positive (metastatic) lymph nodes, while clinical overstaging can be due to subjective clinical assessments falsely interpreted as parametrial spread. Although findings from modern imaging examinations (i.e., ultrasound, MRI, CT, and PET-CT) were not considered for staging purposes in the guidelines until 2018, they had already largely replaced staging results based on conventional procedures in many gynaecologic oncology centres in high-income countries. A multicentric clinical trial by the American College of Radiology Imaging Network/Gynecologic Oncology Group (ACRIN/GOG183) from 2005 showed only sporadic use of conventional procedures recommended for cervical cancer staging [5,7]. Cystoscopy was performed in 8.1%, sigmoidoscopy or proctoscopy in 8.6%, intravenous urography in 1%, and examination under anaesthesia in 27% [5,7]. Similar trends were reported in the European Society of Gynaecological Oncology (ESGO) survey published in 2018; cystoscopy was used during preoperative workup in 17%, rectoscopy in 10%, and examination under anaesthesia in 26% [8]. Therefore, in 2018, the initiative to update the guidelines was undertaken under the joint umbrella of ESGO, the European Society for Radiotherapy and Oncology (ESTRO), and the European Society for Pathology (ESP) and new guidelines for the staging, treatment, and follow-up of cervical cancer patients were established, implementing imaging into the staging and treatment decision-making process [9]. Of note, the updated recommendations after five years (2023) are unchanged in terms of recommended imaging modalities for local, nodal, and distant staging of this disease [10].

## 2. Revised FIGO (2018) and TNM (2021) Cervical Cancer Staging

Precise cancer staging plays a pivotal role in determining prognosis, individualised treatment, and follow-up planning; it also allows for evaluation and comparison of treatment outcomes, rapid and standardised communication between health professionals, and identification of clinical trials for each individual patient. The 2021 version of the American Joint Committee on Cancer (9 AJCC TNM) cervical cancer staging [11] is aligned with the revised 2018 FIGO (the International Federation of Gynecology and Obstetrics) staging for cervical cancer [12,13,14,15,16] (Table 1 Revised FIGO 2018 and TNM staging of cervical cancer and the role of imaging). The most relevant changes between FIGO 2009 and 2018 are presented in Appendix A Key updates to the 2018 FIGO classification compared to the prior version: (1) the incorporation of imaging- and pathology results in the TNM categories and FIGO stage; (2) the division of epithelial tumors based on their association (or lack of thereof) with human papillomavirus infection (WHO 2020); (3) the removal of horizontal dimension as a parameter for T1a (FIGO IA) [17]; (4) the change of size criteria to define T1b including addition of subcategory T1b3 (T1b1 (IB1) ≤2 cm, T1b2 (IB2) >2–≤4 cm, and T1b3 (IB3) >4 cm in maximum tumor diameter); (5) the introduction of pelvic lymph node involvement as N1 (FIGO IIIC1) and para-aortic lymph node involvement as N2 (FIGO IIIC2).

The TNM (T-primary Tumour, N-regional lymph Node, M-distant Metastasis) classification allows a more accurate prediction of disease prognosis than the FIGO stage. For example, if metastases are found in the pelvic and/or para-aortic lymph nodes, patients are considered to have the FIGO stage IIIC disease regardless of the primary tumour size or local pelvic spread [18,19,20]. However, the FIGO stage IIIC group is very heterogeneous and exhibits different survival trends. Therefore, the panelists of ESGO-ESTRO-ESP guidelines agreed that the TNM classification should always be used in cervical cancer patients’ staging [11] and assessed by a multidisciplinary consensus based on the most accurate modality integrating physical examination, imaging, and pathology. Pathological findings supersede imaging and clinical findings. When in doubt, the lower staging should be assigned. The FIGO stage should also be documented since both the FIGO- and TNM stages provide complementary information and are, to some extent, correlated [9,13,15]. The final FIGO stage is first established after all three categories of the TNM staging system are generated. The disease stage is not to be altered later, for example, at recurrence.

A structured checklist is essential for preoperative imaging to ensure appropriate stage assignment and identifying prognostic features relevant to individual treatment planning. It should contain the maximum tumour diameter and, if the reproductive desire is yet to be fulfilled, the shortest distance from the upper edge of the tumour to the internal cervical os and the craniocaudal length of the unaffected cervix [21]. Other findings to be reported are the minimum thickness of the unaffected stroma (tumor-free distance) [22,23]; invasion into the ventral-, lateral- or dorsal parametria; invasion of the vagina (dividing it into upper two-thirds and lower one-third); hydronephrosis (related or unrelated to the extent of the tumour); pelvic side wall invasion into pelvic muscles/fascia/neurovascular structures or the bony pelvis; bladder/rectal invasion (specifying wall and mucosa/lumen invasion); enlarged/suspicious lymph nodes (pelvic and paraaortic or others); adnexal mass(es); other spread (such as peritoneal carcinomatosis, visceral organs parenchymal metastases, etc.); associated benign findings; note the presence of anatomical variants and possible tumour-related complications (e.g., thromboembolism, etc.) [24,25]. The TNM and FIGO staging system is established at the end of the checklist. An example of an ultrasound checklist is presented in Appendix A. The ultrasound checklist on cervical cancer presents a schematic documentation of cervical cancer staging by ultrasound. A video demonstrating ultrasound systematic staging of cervical cancer is available at the link ESGO eAcademy by European Society of Gynaecological Oncology (ESGO) (Accessed on 12 January 2024)

## 3. Local (Pelvic) Workup for Different Stages

The role of modern imaging in local staging is to delineate the cervical tumour to determine if fertility-sparing surgery can be offered, tailor the radicality of parametrial resection based on the minimum thickness of uninvolved cervical stroma, and to assess parametrial infiltration and tumour invasion into the pelvic side wall or adjacent organs (bladder, rectum). Following the revised FIGO 2018 staging system for cervical cancer [13,15], imaging methods include ultrasound, computed tomography (CT), magnetic resonance imaging (MRI), positron emission tomography (PET), combined PET-CT or PET-MRI, etc., based on local resources [12]. In the ESGO survey published in 2018, CT, PET-CT, MRI, and ultrasound were frequently used in the pre-treatment diagnostic workup. The survey showed that in early-stage disease, MRI was the most frequently used imaging method (74%), but more than half of the respondents used CT (54%), and a minority preferred PET-CT (25%). Pelvic ultrasound was reportedly considered in 23% [8]. Real-life data on gynecologic oncologists’ preferred primary staging modality and their diagnostic performance in early-stage cervical cancer was published in the prospective, international SENTIX study [26]. Each participating site was instructed to choose their preferred method based on their routine clinical practice. Among 690 prospectively enrolled patients with early-stage cervical cancer, 46.7% and 43.2% of patients underwent MRI and pelvic ultrasound, respectively, whereas 10.1% underwent both modalities. Pelvic MRI and ultrasound yielded similar diagnostic performance for predicting histological tumour size, parametrial involvement, and macrometastatic nodal involvement. CT is a well-established imaging method being widely used in cancer staging. Improvements in CT technologies using helical and multi-detector CT (MDCT) yield higher spatial resolution with shorter acquisition times, albeit with reported slightly higher radiation exposure [27]. However, CT is still inferior to MRI in assessing tumour size and local tumour extension due to its lower soft-tissue contrast resolution, even when using contrast-enhanced (CE) CT [4,5,28,29,30]. Iodinated contrast agents are routinely used for CT examinations to visualise contrast-enhancing neoplastic lesions or metastases and diagnose various non-malignant conditions, e.g., infectious- and vascular disease (Figure 1) [4,28]. PET-CT provides a unique combination of anatomic information provided by CT and tissue-specific metabolic characteristics provided by PET using the glucose analogue (18)F-fluorodeoxyglucose (FDG). The fused images acquired during a single examination facilitate localising malignant lesions, typically exhibiting increased FDG-avidity and depiction of physiologic FDG uptake in non-malignant tissue. However, PET-CT is not optimal for local staging due to the low soft-tissue contrast on CT and the low spatial resolution of PET. Furthermore, the partial volume effect from FDG-avid urine in the bladder (physiologic renal FDG excretion) may preclude an accurate definition of the tumour volume or parametrial invasion (Figure 1) [31]. PET-CT involves slightly higher radiation exposure than diagnostic CT alone [32]. MRI thoroughly assesses the pelvic anatomy with a wide field of view and no radiation risk. MRI has for years been considered the modality of choice for detecting local tumour spread [33] and has been shown to yield high interobserver reproducibility for tumour size measurements with high concordance between maximum primary tumour size from MRI and from hysterectomy specimens [34]. However, MRI is relatively expensive and time-consuming and may be contraindicated in some patients (e.g., in the presence of MRI-incompatible implants). Furthermore, limitations in access to MRI scanners are particularly common in low-income countries. As for all imaging modalities, their diagnostic accuracy depends on the radiologist’s experience in gynaecologic oncologic imaging. The MRI protocol, traditionally based on morphological sequences, has recently been supplemented by functional sequences, including diffusion-weighted imaging (DWI) and dynamic contrast-enhanced (DCE) MRI (Figure 1) [35,36]. DWI depicts the free water motion of the tissue. The free water movement is restricted in malignant lesions, normally being highly cellular. The tumour appears hyperintense on high b-value (e.g., b = 1000 s/mm^2^) images and correspondingly hypointense on the apparent diffusion coefficient (ADC) maps. The ADC map depicts true restricted diffusion and allows measurements of tumour ADC value. DWI combined with conventional MRI sequences enable assessments of both morphologic and physiologic features in a single examination. In DCE-MRI, dynamic image acquisition is accomplished after the administration of an intravenous bolus of gadolinium-based contrast agent. Typically, cervical tumours enhance rapidly, followed by an early washout of contrast. In the early arterial phase (30 sec post-contrast), the tumour is hyperintense. In contrast, in the late venous phase (2 min post-contrast), it is hypointense relative to the more gradually enhancing normal cervical epithelium and stroma [29,35]. Using a contrast agent could increase the reader’s confidence in identifying stromal and parametrial invasion. On the other hand, no significant improvement in staging accuracy has been demonstrated, and therefore, its use is not considered essential [37]. The addition of DWI to T2-weighted MRI demonstrated a promising role in improving the detection of parametrial invasion and increasing reader confidence [38,39], allowing better tumour delineation for less-experienced radiologists. Importantly, the measured maximum tumour dimensions are reportedly virtually identical based on DWI and conventional series. Functional magnetic resonance, including DWI and DCE-MRI imaging, have also been recently addressed and studied as a tool supplementing conventional MRI in brachytherapy settings for patients with locally advanced cervical cancer. Their complementary use resulted in lower interobserver variability in target delineation (Gross tumour volume) [40]. Nevertheless, validation through robust prospective data, before extensive adoption of DWI and DCE-MRI in cervical cancer, is essential. Particular attention must be paid to the use of uniform protocols, standardised nomenclature and correlation of imaging findings with histopathology [41,42]. PET-MRI integration has not yet been shown to significantly improve local staging performance compared to MRI alone [43]. Examination from the renal hila to the pubic symphysis is recommended to assess the presence of hydronephrosis in case of pelvic wall and lymph node infiltration (see below) [36].

The diagnostic potential of ultrasound is likely to have been underestimated in gynaecologic oncology until recently. In cervical cancer, ultrasound was formerly reserved for the screening of hydronephrosis. The reported limitations of ultrasound were low-contrast resolution, which may have limited the differentiation of the tumour from the adjacent tissue, small field of view disallowing the evaluation of the pelvic side wall, dependence on operator skill, subjectivity when interpreting the image, and challenging technical storage and retrieval of high-quality images on demand. However, ultrasound has undergone major technological developments over the past decades, especially the development of endovaginal high-resolution probes with a wide field of view, allowing the depiction of detailed pelvic anatomy comparable to that from MRI (Figure 1).

Ultrasound can be performed by gynaecological oncologists who benefit from the meticulous knowledge of the disease. The same endoluminal probe can be introduced transvaginally or transrectally. The transrectal approach, performed without any patient preparation, such as enema or fasting, is preferred for cervical cancer due to the diminished risk of bleeding from the tumour. Additionally, the transrectal approach allows a better acoustic setting to show the distal part of the cervix [44]. The combination of transvaginal/transrectal and transabdominal ultrasound allows a complete assessment of the abdomen and pelvis for abdominal staging (Figure 2) [45].

Nowadays, expert sonographers dedicated to gynaecologic oncology scanning can perform local staging by evaluating the maximum tumour size, depth of stromal invasion, infiltration of pericervical fascia and parametrial involvement up to the pelvic side wall, including the assessment of iliac vessels, pelvic muscles and nerves, and the depth of bladder and rectal wall infiltration (Appendix A Ultrasound checklist on cervical cancer) [46,47,48,49,50]. Moreover, the dynamic aspect of an ultrasound scan can be applied to reliably exclude the infiltration of adjacent organs (bladder and rectosigmoid colon) [51,52]. In addition to two-dimensional (2D) ultrasound, three-dimensional (3D) ultrasound obtains the third (coronal) plane, similar to MRI. It enables easy storage of measured volumes and their retrieval for second readings, planning radiotherapy, and evaluating the treatment effect (Figure 3). Moreover, Doppler ultrasound allows highly accurate, non-invasive, in vivo assessment of vascular features reflecting tumour angiogenesis, which has been shown to predict clinical response to neoadjuvant or definitive chemoradiation in patients with locally advanced cervical cancer [53,54]. The transabdominal ultrasound not only evaluates hydronephrosis [55] but may also detect abdominal intraparenchymatous metastases, lymph node metastases, and peritoneal spread during one examination (see below) [45]. The ultrasound examination is well tolerated by patients and does not require any patient preparation, such as fasting or contrast agent application. Contrast-enhanced ultrasound is not established in cervical cancer staging [56,57]. In addition, ultrasound is widely available and cheaper than MRI and does not have any known contraindications (Table 2 Comparison of Different Imaging Methods for Application in Gynecologic Oncology).

The introduction of ultrasound into cervical cancer staging depends on the availability of a trained sonographer. Whereas MRI could acquire images without needing an on-site MRI specialist with subsequent readings by an experienced radiologist. Another limitation may be the lack of routine standardised storage of ultrasound images, which makes it impossible to present ultrasound images at multidisciplinary meetings. ESGO, the International Society of Ultrasound in Obstetrics and Gynecology (ISUOG), FIGO and other scientific societies recognise the role of ultrasound in gynecologic oncology, particularly for cervical cancer in low-resource countries where the incidence is highest and availability of imaging is limited. ESGO and ISUOG have recently introduced training in gynaecological-oncology imaging as part of their curricula. An experienced sonographer should document the extent of the disease using a predefined checklist and high-quality dynamic and 3D images with easy on-demand retrieval. There are no acoustic limitations to pelvic scanning, such as obesity or intestinal gas, which may limit abdominal retroperitoneal staging in some patients (Table 2).

Acknowledging some of the advantages of ultrasound over MRI (Table 2 Comparison of Different Imaging Methods for Application in Gynecologic Oncology), the next step before implementing ultrasound as a possible first-choice procedure for pelvic staging is to demonstrate its comparable diagnostic performance with that of MRI, in the same patient cohort. The diagnostic performance of imaging is ideally assessed using histopathology as the reference standard. Full-section tumour specimens are easily available in early-stage cervical cancer undergoing definitive surgical treatment. Thus, evidence for the role of ultrasound in comparison to state-of-the-art imaging modality (MRI) in local staging of early-stage cervical cancer can be retrieved from recent European multicentric prospective trials comparing both imaging modalities in the same cohort, the results of which are presented in the text below [48]. Whereas in patients with locally advanced cervical cancer who are subjected to definitive chemoradiation, histological confirmation of local infiltration is missing. In lack of a true reference standard, some authors have tried to compare ultrasound to MRI using MRI as a reference standard [44], although studies have reported MRI to yield similar or lower accuracy than ultrasound for staging in early-stage cervical cancer [48]. Although MRI is significantly better than CT (*p* = 0.047) for detecting parametrial invasion with reported high specificity (77–80%) and negative predictive value (83–87%), the reported sensitivity (40–57%) and positive predictive value (32–39%) of MRI for detecting incipient parametrial spread are still low [58]. Another important aspect of the implementation of the imaging method of choice in primary work-up is to prove their inter-observer reproducibility [59,60]. A multicentric trial comparing the inter-observer agreement of transvaginal ultrasound and MRI in the assessment of local tumour extension in cervical cancer patients in relation to observer experience showed that the inter-observer agreement was moderate for ultrasound (κ values 0.41–0.60), and moderate-substantial (κ values 0.61–0.80), for MRI. The experience of the ultrasound examiners was associated with inter-observer agreement only for parametrial invasion [59]. The results emphasise the importance of systematic training of the sonographers performing cervical cancer staging [59]. Interestingly, no difference in agreement for staging parameters among radiologists with different levels of pelvic MRI experience was reported in a recent cervical cancer MRI patient cohort (n = 416) [61].

Data concerning the local tumour staging using the recommended imaging methods is given in the following steps, from tumour detection and delineation to assessment of its extent in the surrounding organs:

### 3.1. Tumour Detection

The first step in local staging by ultrasound or MRI is the identification of cervical cancer tissue, which is assessed in relation to the surrounding cervical stroma. On ultrasound, a squamous cell cancer is characterised by a hypoechogenic, richly vascularised tumour, while adenocarcinoma is more often an iso- or hyperechogenic, highly perfused lesion contrasting with healthy residual cervical stroma (Figure 4) [62]. On T2-weighted MRI, cervical cancer is typically an iso- or hyperintense mass regardless of histopathologic type, and when small, is surrounded by a hypointense normal cervical stroma (Figure 4) [63]. Tumour hyperintensity on the high b-value DWI with corresponding low intensity on the ADC map represents true restricted diffusion in the lesion. Importantly, the inclusion of DWI in addition to T2-weighted imaging in cervical cancer has been shown to improve tumour detection and reader confidence and yield better diagnostic accuracies for predicting parametrial invasion [39,64]. Promising results on early-stage tumour detection by ultrasound versus MRI based on single-unit studies [46,47] have been reported by a European multicentre trial including 189 patients operated for early-stage cervical cancer [48]. The transvaginal or transrectal ultrasound/MRI (based on T2-weighted and T1-weighted series without DWI) yielded accuracies (sensitivities) [specificities] of 96%/86%, *p* < 0.001 (90%/67%, *p* = 0.008) [97%/89%, *p* = 0.005] for tumour detection. This study found an excellent agreement between ultrasound and final histology for detecting tumours (kappa value 0.84), while the agreement between MRI and final histology was only moderate (0.52).

### 3.2. Tumour Delineation within Cervix (Tumour Size, Depth of Stromal Invasion, Minimum of Uninvolved Stroma, and Cranial Tumour-Free Margin)

The next step is to delineate tumour borders within the cervix to assess important prognostic parameters (tumour size, depth of stromal invasion, minimum of uninvolved stroma, and cranial tumour-free margin and others), which guide the treatment options and enable individualised management, including fertility-sparing treatment (FST). To assess eligibility for FST, the joint ESGO-ESTRO-ESP cervical cancer guidelines recommend ultrasound and MRI as the imaging tests of choice to measure remaining cervical length (after cone biopsy), uninvolved cranial tumour-free margin, and residual tumour size (Figure 4) [9]. FST is only feasible in squamous cell cancer or HPV-associated adenocarcinoma with a largest diameter of less than or equal to 2 cm [9]. A multicentric European trial focused on the tumour detection rate after cone biopsy, comparing the diagnostic performance of ultrasound and MRI (without DWI) [48]. The residual tumour detection rate after cone biopsy was not significantly different from tumour detection rate after cervical biopsy alone if assessed on ultrasound (sensitivity: *p* = 1.0; specificity: *p* = 0.23) or MRI (sensitivity: *p* = 0.78; specificity: *p* = 0.25) [48]. The study reported good agreement between ultrasound and MRI in classifying small tumours less than 2 cm (kappa values 0.78 and 0.71, respectively). For FST planning, the most important parameter to be evaluated is the actual tumour-free distance between the cranial tumour margin and the internal os (Figure 4). A recent meta-analysis published by Xiao et al. in 2020 on the diagnostic performance of MRI in evaluating the distance between the tumour and the internal os on six studies (454 patients) showed a pooled sensitivity and specificity of 87 and 91%, respectively [65]. A meta-analysis published by Woo et al. in 2020 analysed five MRI studies and showed a pooled sensitivity of 84% and specificity of 96% in detecting internal os involvement [66]. A multi-institutional prospective study compared the accuracy of ultrasound and MRI for tumour detection, tumour size measurements, parametrial-, uterine corpus- and vaginal fornix involvement and prediction of FIGO/T-status (from TNM system), reporting no significant difference between the two imaging methods [67]. Regarding assessment of cervical internal os invasion and uterine corpus infiltration as a negative prognostic factor, the accuracy for ultrasound and MRI was 94% and 86%, respectively (*p* = 0.3) [67]. On top of that, ultrasound can also be used to guide the surgeon intraoperatively during FST to determine the optimal level of excision in order to preserve the maximum length of tumour-free cervix for a future pregnancy [68]. In the pilot study, the cranial tumour-free margin was marked intraoperatively under ultrasound guidance with a non-absorbable suture [68]. The pathology report showed a mean distance between the stitch and the cranial tumour margin of 1.5 mm (SD 1.16, range 0.09–5 mm) [68]. In cases in which neoadjuvant chemotherapy (NACT) precedes FST with the aim to achieve partial or complete tumour response with sufficient cranial tumour-free margin, MRI or ultrasound in experienced hands can also be used to assess the treatment response [69]. In a single-unit prospective study comparing MRI and ultrasound after NACT in 42 patients using pathological results as a reference, the agreement between measurements obtained by MRI (without DWI) and histology was not found statistically significant (intraclass correlation coefficient; 0.344; 95% CI: −0.013 to 0.610; *p* = 0.059), while agreement between transrectal ultrasound and histology reached statistical significance (intraclass correlation coefficient; 0.795; 95% CI: 0.569–0.902; *p* < 0.001) [69]. Individualised planning of care is not only limited to FST but also to assess the radicality of hysterectomy or primary chemoradiotherapy, depending on the tumour size and depth of stromal invasion. Tumour size >4 cm and deep stromal invasion are indicative of a worse prognosis; thus, they are used in many centres as an indication for primary chemoradiation instead of primary surgery. The multicentric European trial results demonstrated almost perfect agreement between ultrasound and histology in the assessment of such bulky tumours (>4 cm) and deep stromal invasion (kappa values 0.82 and 0.81, respectively) [48]. The agreement between MRI (without DWI) and histology was substantial for the classification of bulky tumours (>4 cm) and detection of deep stromal invasion (kappa values 0.76 and 0.77, respectively) [48]. Recently, the measurement of the distance between tumour and parametria (tumour-free distance or a minimum uninvolved stroma) has been recommended as it better correlates with the risk of extrauterine extension and nodal metastasis rate than the tumour size or depth of stromal invasion, which does not consider the size of the cervix and the tumour location within the cervix [22,23]. It is measured as the remaining uninvolved fibromuscular stroma between the tumour and pericervical fascia at the point where the ventral, lateral, and dorsal parametria are attached to the cervix. The cut-off values for tumour-free distance associated with clinical outcomes ranged between 2.5 and 3.5 mm but without prospective validation [22,23]. Looking at the available evidence, pre-surgical MRI showed a sensitivity of 88% and a specificity of 75% in the assessment of tumour-free distance [70].

### 3.3. Extrauterine Extension (Vagina, Parametria, Pelvic Side Wall, Hydronephrosis and Others)

The third step is the assessment of extrauterine extension. The vaginal extension is routinely assessed during physical examination. The estimation of vaginal fornix using imaging can be difficult especially in large tumours stretching the vaginal fornix. An optional, useful tool to better assess the vaginal extension of the tumours is represented by vaginal opacification with gel, especially when the region of interest is represented by the posterior vaginal fornix. On the other hand, the role of imaging is critical to assess pericervical fascia and parametrial involvement, including pelvic side wall invasion. Visualisation of intact pericervical fascia surrounding the cervix as a hyperechogenic line on ultrasound or a hypointense line on MRI excludes infiltration of parametria with specificity and a negative predictive value of 98–100% [46,71,72]. In addition, the dynamic aspect of ultrasound examination helps to establish parametrial status. Especially in situations with limited visibility, the exertion of sliding of the tumour against the surrounding tissue planes is a sign of intact parametria. A multicentric European trial of early-stage cervical cancer of transvaginal or transrectal ultrasound/MRI (based on T2-weighted and T1-weighted series without DWI) yielded accuracies (sensitivities) [specificities] of 97%/90%, *p* = 0.001 (77%/69%, *p* = 0.56) [98%/92%, *p* < 0.001] in the assessment of parametrial invasion [48]. A recent meta-analysis published by Alcázar et al. in 2020 reported similar diagnostic performance for detecting parametrial invasion in cervical cancer by ultrasound/MRI, with pooled sensitivities and specificities of 78%/68% and 96%/91%, respectively [73]. No statistical differences were found when comparing both methods (*p* = 0.548) [73]. These data were confirmed by another meta-analysis published in 2020 by Woo et al. and showed that ultrasound has a comparable level of diagnostic performance to MRI in assessing parametrial invasion (pooled sensitivities and specificities of 67%/71% and 94%/91%, respectively) [66]. Apart from establishing the involvement of parametria, the precise localisation of infiltration (ventral right/left, lateral right/left, dorsal right/left) represents an added benefit for radiotherapy planning (Figure 1 and Figure 3) [44]. A narrated video by Moro et al. guided the reader through the methodology of assessing the structures surrounding the cervix and vagina (specifically the parametrium) [74]. Chiappa et al. compared the agreement between 2D and 3D ultrasound to MRI results as a reference in assessing parametrial invasion and showed the best agreement in the assessment of ventral parametria (90% and 62.5%), followed by the right lateral parametrium (72% and 81%), left lateral parametrium (69% and 70%), and dorsal parametria (58.5% and 52%) [44]. Based on the study results, 2D- and 3D-ultrasound showed similar moderate agreement with MRI. In addition to the location of infiltrated parametria, the degree of parametrial invasion can also be assessed on ultrasound or MRI using a standardised grading system, which is crucial for the treatment choice [45]. The features of parametrial invasion are incipient infiltration of pericervical fascia (usually in depth ≤5 mm), grade 2; nodular infiltration of parametrium, grade 3, discontinual parametrial involvement (“skip-metastasis”), grade 4 [45]. Metastatic visceral paracervical lymph nodes are considered discontinual parametrial invasion on ultrasound (grade 4) [46]. TNM and FIGO do not define how to classify metastases in the para-uterine visceral lymph nodes [11,15]. Since the lymphatic spread from cervical cancer is initially to these visceral (parametrial) lymph nodes drained by internal iliac vessels (e.g., uterine vessels), their involvement could be classified as locoregional lymph node metastases and not as infiltrated parametria. Parametrial invasion towards the pelvic side wall upstages the disease from T2b to T3b [11,15]. The pelvic side wall is defined as the parietal muscles of the lesser pelvis (obturator internus, coccygeus, and piriformis muscle), fascia, neurovascular structures, or skeletal portions of the bony pelvis. Pelvic side wall invasion by a tumour is a frequent cause of ureteric obstruction associated with hydronephrosis (Figure 5). Additional use of transabdominal ultrasound is essential for screening of hydronephrosis with a sensitivity of 76.5%, specificity and positive predictive value of 100%, negative predictive value of 85%, and accuracy of 90% [55]. The location of ureteric obstruction can be easily identified using a combination of transabdominal and transvaginal/transrectal ultrasound or MRI following the visualisation of suprastenotic dilatation of the ureter. Hydronephrosis on MRI can be examined if the protocol includes a sequence with an extended field of view to both kidneys in the axial or coronal plane. The degree of hydronephrosis is divided into three grades, as has been previously described [45].

### 3.4. Extension to Surrounding Organs (Bladder, Rectum, Sigmoid Colon)

The last step in local staging focuses on the assessment of tumour growth into the adjacent organs (bladder and rectum or sigmoid colon). The level of infiltration of both the bladder and rectum can be determined simultaneously using an identical grading system (see below) [45]. Ultrasound or MRI are used to detect the infiltration of the endopelvic fascia based on the assessment of the contact planes between adjacent organs and the extent of involvement of both the bladder and rectal wall (Figure 6 and Figure 7) [45,52].

In a single-unit study by Iwamoto et al., the accuracy of ultrasound in the detection of bladder infiltration was superior (95%) to other methods (76% for CT, 86% for cystoscopy, and 80% for MRI) [51]. It has been proposed that the better results achieved with ultrasound are related to the dynamic aspect of the ultrasound examination; thanks to the positive sliding effect between organs, the authors could exclude bladder wall infiltration. The same manoeuvre can be used to exclude rectosigmoid wall infiltration. Huang et al. described ultrasound characteristics representative of the different stages of bladder wall invasion [52]. In 2011, Fischerova categorised the infiltration of the bladder wall on ultrasound examination using four stages (0–3), from uninterrupted endopelvic fascia to complete disruption of the hyperechogenic mucosa with the presence of intraluminal tumour nodules [45]. The same grading system can be used to assess the depth of rectal or sigmoid wall involvement [45].

Ultrasound and MRI detect sequential changes in all the layers of the bladder or rectosigmoid wall with high accuracy, offering more information on the disease burden than the endoscopic examination (cystoscopy, rectoscopy), which only depicts the worst stage of infiltration (infiltration of the mucosa) [52,75,76]. In a meta-analysis published in 2020 by Woo et al., the pooled sensitivity and specificity of five studies using MRI in bladder wall infiltration assessment were 84 and 95%, respectively [66]. The minimally invasive diagnostic techniques, cystoscopy or rectoscopy, can be considered redundant because the results of these examinations do not change the management and only delay the treatment initiation. For that reason, the joint ESGO-ESTRO-ESP cervical cancer guidelines recommend cystoscopy and rectoscopy only for obtaining a biopsy in cases of suspected synchronous secondary malignancy [9].

To sum up the local workup and present literature, the reported slightly better accuracy of ultrasound over MRI to delineate the tumour may be due to the combination of sonomorphology corresponding to features of different histological subtypes and evaluation of tumour vessels by Doppler defining the extent of the tumour growth within the cervix, as well as the use of suboptimal MRI protocols (without DWI) according to current imaging guidelines [36]. The slightly better results for ultrasound in comparison to MRI (also without DWI) in parametrial invasion assessment may be firstly due to the recent technical improvements in ultrasound technology, including high-frequency endoluminal probes enabling detailed visualisation of the cervix contour and pericervical fascia. Secondly, it may be due to the Doppler scan’s discriminating accurately between cervical vessels and tumour spiculae into the parametria. Thirdly, it may be related to the possibility of performing dynamic evaluation during the ultrasound examination. Using a slight pressure of the probe against the cervix, the ultrasound expert can easily observe the sliding of the tumour against the surrounding tissue (positive sliding sign) and exclude their infiltration. Lastly, based on the excellent tissue plane resolution and dynamic aspect of examination, ultrasound excludes or confirms the infiltration of the bladder or rectum and precisely assesses the tumour depth of infiltration in the bladder/rectal wall. In accordance with the current evidence, the joint ESGO-ESTRO-ESP cervical cancer guidelines accepted transvaginal/transrectal ultrasound (TRS/TVS) as an effective alternative to the pelvic magnetic resonance (T2WI and DWI-MRI) in the primary workup of cervical cancer [9,10]. Moreover, the revised FIGO staging emphasised that ultrasound in the hands of experienced operators can provide comparable information to MRI for the staging of cervical cancer [15].

## 4. Nodal and Distant Diagnostic Workup

Regarding studies on the diagnostic performance of different imaging methods in the detection of metastatic lymph nodes, bias has stemmed from heterogenous study populations (early, advanced stage), discrepant study designs (retrospective vs. prospective), different levels of technical equipment, accuracy calculated per patient instead of per node or region, the wide time range of studies selected in meta-analysis (which may lead to the inclusion of obsolete/limited technologies), absence of prospective comparison of different imaging methods in one cohort and others. All these limitations lead to heterogeneous data and, therefore, difficulties in drawing meaningful conclusions. For this reason, a multicentre prospective study comparing ultrasound, DWI-MRI and FDG PET-CT in nodal staging has been initiated, and the results are expected in 2025 (CANNES study).

On ultrasound, the combination of endoluminal (transvaginal/transrectal approach), convex (transabdominal approach) and linear array probes (transcutaneous approach) are used to detect regional (pelvic [parailiac] and abdominal [paraaortic]) and distant nodal involvement (supraclavicular, inguinal, axillar and others) (Figure 8).

Instead of size, the integration of sonomorphological and vascular features of lymph nodes should be considered to differentiate metastatic infiltration from normal nodes or those with reactive changes. A consensus from the VITA (vulvar international tumour analysis) group was published in 2021, describing the lymph nodes on grayscale and Doppler ultrasound as normal, benign (reactive, post-reactive) or infiltrated (Figure 9) [77]. In an infiltrated node, the shape is usually round due to the tumour load, there is a loss of the nodal core sign (a functional unit formed by the nodal medulla with or without hilum), and there is inhomogeneous echogenicity caused by cystic necrosis and calcifications. Metastatic lymph nodes in cervical cancer are usually markedly hypoechogenic. They can show capsular interruption as a marker of extracapsular growth, with irregular margins and diffuse infiltrating growth into the vessels and surrounding tissues. Hilar flow may still be preserved in a partial nodal involvement with or without transcapsular vascularisation (vessels penetrating the cortex from outside) the latter are usually found in an advanced stage of infiltration (Figure 9 and Figure 10) [45,77].

According to the VITA consensus, lymph nodes are characterised as LN1, normal lymph node (exhibiting normal morphology); LN2, benign (showing reactive or post-reactive lymph node changes); LN3, indeterminate but probably benign (manifesting some features that preclude classifying the lymph node as certainly benign); LN4, probably malignant (showing eccentric cortical thickening, suggestive of intranodal metastasis); and LN5, malignant (exhibiting frank malignant features like spiculated shape, capsular interruption, perinodal hyperechogenic ring, absence of nodal-core sign, intranodal cystic areas or reticulations and transcapsular vessels (Appendix A Ultrasound checklist on cervical cancer) [77].

Similarly to ultrasound, CT and MRI traditionally evaluate lymph nodes based on their size and morphological criteria. Metastatic lymph nodes often show a round shape, irregular margins, inhomogeneity, loss of fatty hilum, and a tendency to increase enhancement after intravenous contrast administration. Recently, it has been shown that using DWI depicting restricted diffusion in metastatic lymph nodes is helpful for detecting lymph node metastases (Figure 10) [78]. Since it is not exposed to radiation, DWI/MRI is clearly preferred in the staging of pregnant women with cervical cancer [79]. An advantage of PET-CT is the fusion of anatomical information provided by CT and metabolic information provided by PET [80]. FDG PET-CT can detect metastatic spread even in normal-sized nodes by demonstrating high tumour-related metabolic activity (Figure 10). The ability of FDG PET-CT to correctly identify metastatic lymph nodes is, however, largely affected by lymph node size, with node-based reported sensitivities of 100%, 67%, and 13% in metastatic nodes ≥ 10 mm, 5–9 mm, and ≤4 mm, respectively, in uterine cancers [81]. PET-CT may also play an important role in the correct identification of lymph nodes enlarged due to benign reasons. Novel imaging techniques such as FDG (fluorodeoxyglucose)- hybrid PET-MRI integrates high-resolution multi-planar morphologic and functional information from MRI with the metabolic data from FDG PET simultaneously (Figure 11). As opposed to CT, MRI yields superior soft tissue contrast and relatively high spatial resolution and imposes no ionising radiation, while PET offers high sensitivity for metabolic alterations due to the detection of a high metabolism, which seems to be useful for differentiating between metastatic and benign lymph nodes and may reduce false findings [82]. Recent data suggest that FDG PET-MRI is equivalent to MRI and superior to FDG PET-CT for local staging of the primary tumour, and FDG PET-MRI is comparable to FDG PET-CT for nodal staging [43,83]. PET-MRI is more expensive than PET-CT as it combines the cost of a full MRI with a full PET scanner and requires the increased imaging times of MRI compared to CT. However, in patients who require both MRI and PET, the overall scanning time will be reduced with improved localisation of the PET signal. Fused images from PET-MRI might improve the staging results, which will need to be evaluated in a large, multicentric, prospective study.

The detection rate of nodal involvement and distant metastases depends on the tumour stage (prevalence of metastatic lymph nodes) and the size of metastatic lesions. Imaging (ultrasound, MRI, CT or PET-CT) shows high specificity in the detection of nodal metastases (>90%) but very low sensitivity in the detection of micrometastases and small-volume metastases (52%, 38%, 52%, and 54%, respectively) [84,85]. In the pelvis, occult nodal metastases are usually detected during surgical staging and/or if chemoradiation is indicated instead of primary surgery, they are routinely included in the radiation field. The crucial point is to detect lymph node metastases outside the routinely applied radiation field in locally advanced cervical cancer. In patients with locally advanced cervical cancer at the time of initial management, the presence of extrapelvic spread, particularly in the paraaortic nodes and/or chest, ranges from 10% to 30% [86]. Missing occult disease on the conventional imaging contributes to persistent nodal and/or distant involvement that eventually progresses, often with fatal outcomes.

In accordance with European guidelines, ultrasound or MRI is mandatory in the primary work-up and local staging of cervical cancer [10]. While assessing the local stage in the pelvis, ultrasound and MRI both also screen for pelvic metastatic lymph nodes during the same evaluation. There are still less available data for the diagnostic performance of ultrasound than MRI, but looking at real-life data retrieved from the SENTIX study, among 690 patients, ultrasound (n = 298) is similarly represented to MRI (n = 322) in the primary workup assessing cervical cancer extent in the pelvis including the pelvic lymph nodes with comparable results between both methods [26].

In the early stages (T1a, T1b1, T1b2, T2a1), micrometastases are often undetected, and the sensitivity of imaging is generally low. Traditional size criteria to distinguish between benign and metastatic lymph nodes for MRI and CT have a cut-off for a short axis diameter of 10 mm. However, using this size cut-off yielded relatively low sensitivity (30–73%) [78]. Similarly, the negative prediction value of PET-CT in lower tumour stages was not sufficient, and micrometastases were often undetected (cut-off for malignant tissue detection is 5 mm) [87,88,89]. A retrospective analysis by Frühauf et al., from a prospective registry of one specialised cancer centre, considered 390 patients examined on preoperative ultrasound with available surgical staging. In this study, the median largest diameter did not differ between metastatic and non-involved pelvic nodes, reaching 10.0 mm in both groups. The pelvic intranodal metastases exhibited a median largest diameter of only 4.0 mm. The performance of ultrasound in detecting lymph node metastases was substantially different for early stages and locally advanced cervical cancers, showing a sensitivity of 9.5% and 70.4%, and a specificity of 98.4% and 78.3%, respectively [90]. In the large prospective multicentric SENTIX study, preoperatively on ultrasound or MRI, unrecognised pelvic lymph node involvement was detected by sentinel lymph node mapping in 68 (9.9%) patients, of which 54.4% and 45.6% had micrometastasis and macrometastasis, respectively. Ultrasound and MRI were comparable in the accuracy of detection of sentinel lymph node macrometastasis (*p* = 0.868) [26]. A meta-analysis by Selman found that, in determining lymph node status, sentinel node biopsy reached a pooled positive likelihood ratio of 40.8, with a pooled negative likelihood ratio of 0.18 which makes the sentinel lymph node mapping an almost ideal diagnostic test. Sentinel node biopsy was found to be 20 times more accurate than MRI and five times more accurate than PET-CT, according to the odds ratio scale [91]. Sentinel node biopsy is a less invasive and more accurate assessment of lymph node status and should be performed as the first step of surgical assessment in this cohort of patients. Only in the T1a1 stage without lymphovascular invasion, the incidence of lymph node metastases is extremely low, and consequently, lymph node staging is not recommended [9].

In locally advanced cervical cancer (T1b3 and higher, except T2a1), several meta-analyses have been published on the diagnostic performance of conventional and functional imaging modalities in lymph node staging (Appendix A: Overview of meta-analyses on radiological local and nodal staging in cervical cancer). Although their methodology is of good quality, the very heterogenous data makes conclusions regarding the best diagnostic method unreliable, particularly on the standard use of PET-CT in the primary workup. The outcomes of imaging methods strongly depend on the ratio of early and locally advanced stages of included cancers, strict adherence to pathology as the reference standard and proportion of SLN biopsies with pathological ultrastaging. All published meta-analyses on the diagnostic performance of preoperative imaging should be considered with this caution [30,66,78,84,92,93,94]. In the largest retrospective single-unit study, including 390 consecutive patients enrolled between 2009 and 2019 with cervical cancer and pathological verification of lymph nodes, all patients had preoperative ultrasound imaging for staging purposes [90]. The study confirmed that the only significant factors that affected the diagnostic performance of ultrasound in lymph node assessment were related to the local stage of disease. The largest diameter of the tumour ≥40 mm and invasion to parametria were associated with significantly higher sensitivity of nodal involvement detection (67.9% vs. 35.0%, *p* < 0.001; resp. 74.4% vs. 30.6%, *p* < 0.001) [90]. In the study centre, only ultrasound was routinely used for loco-regional staging, thus no comparison with MRI was possible. On the other hand, an ultrasound assessment of the paraoartic region was a mandatory part of the examination protocol. Metastases in paraaortic lymph nodes (macrometastases only) were confirmed in 16 patients out of 71 who underwent paraaortic lymphadenectomy. Ultrasound imaging yielded a sensitivity of 56.3%, a specificity of 98.2%, and an AUC of 0.772 to identify paraaortic lymph node macrometastases [90]. Regarding MRI, including DWI (Figure 10), this imaging modality seems to be useful in differentiating between metastatic and benign lymph nodes [78]. The meta-analysis published by Shen et al. in 2015 evaluated the diagnostic performance of DWI in the detection of pelvic lymph nodes in cervical cancer and documented pooled sensitivity and specificity of 86% and 84%, respectively [78]. The results were confirmed by a meta-analysis by Liu et al. in 2017, and by Woo et al. in 2018 [92,95] Another meta-analysis on the same topic published by Gong et al. in 2017 showed pooled sensitivity (specificity), and area under the curve [AUC] of DWI 84% (95%) [0.95], ultrasound 71% (99%) [0.90], PET-CT 68% (97%) [0.94], PET 56% (97%) [0.96], conventional MRI 50% (95%) [0.80], and CT 47% (93%) [0.74) [96]. The summary of the receiver operating characteristic (ROC) curve indicated a promising role of DWI in the detection of pelvic lymph node metastases. Regarding tumour extension assessment outside the true pelvis, PET-CT shows very high specificity and it is currently recommended in patients treated with curative intent [66,84,86,92,93,96,97,98].Regarding paraaortic regions, five meta-analyses have been published (Appendix A) [66,84,86,93,97]. A meta-analysis by Choi et al. in 2010 concluded that the PET/PET-CT had better diagnostic performance than CT or MRI based on weak data. The meta-analysis did not distinguish between different CT and MRI developmental stages, patient characteristics, and pelvic and paraaortic lymph nodes [84].

In region- or node-based data analysis, pooled sensitivities of CT (52%) and PET-CT (54%) were higher than for MRI (38%), while pooled specificities for MRI (97%) and PET-CT (97%) were higher than CT (92%). However, the area under the curve (AUC) showed no significant difference between CT, MRI, and PET-CT. Another meta-analysis in 2010 initiated by Kang et al. examined the diagnostic quality of PET-CT in the diagnosis of only paraaortic lymph node metastases in patients with cervical carcinoma. The authors concluded that the pooled sensitivity of PET-CT was low, with a wide confidence interval among the studies (34%, 95% CI 10–72%), while the pooled specificity was consistent (97%, 95% CI, 93–99%) [86]. Only in patients with a high probability for metastases with a prevalence of lymph node involvement of more than 15% performed the PET-CT detect paraaortic lymph node metastases with a pooled sensitivity of 73%, false positive rates of 35%, and false negative rates of 5% [86]. The detection of paraaortic lymph node metastasis by PET or PET-CT was also studied as a subgroup analysis in a meta-analysis by Liu et al. in 2017 comparing CT, MRI, PET or PET-CT and DWI-MRI for detecting lymph node metastases in cervical cancer patients [92]. PET-CT showed a pooled sensitivity of 81.0% (95% CI = 0.52–0.95) and a pooled specificity of 98.0% (95% CI, 0.93–0.99), outperforming both CT and MRI; no data on DWI-MRI performance for this subgroup analysis were available. When considering both pelvic and paraaortic lymph node metastasis DWI-MRI outperformed PET-CT in terms of pooled sensitivity being respectively 87.0% (95% CI, 0.82–0.91) and 66.0% (95% CI, 0.56–0.75) [92]. More recently, Yu et al. conducted a meta-analysis, published in 2019, with the same intent of examining PET-CT performance only in paraaortic lymph node metastases, obtaining pooled estimates for sensitivity and specificity of PET-CT 71.0% (95% CI, 0.54–0.83) and 97.0% (95% CI, 0.93–0.98), respectively [97]. Those results were consistent with the meta-analysis by Ruan et al. published in 2018, including both pelvic and paraaortic lymph nodes, although their subgroup analysis of para-aortic lymph node metastasis led to less robust results due to the small size of the selected studies and the mainly retrospective data [93]. Based on the limitations of non-invasive imaging techniques to accurately detect paraaortic lymph node metastases in locally advanced cervical cancer and poor outcomes of unrecognised occult nodal disease localised outside the radiation field, there is a potential role for surgical staging being not only a diagnostic and prognostic tool but also a therapeutic modality to remove the bulk of the disease [99,100,101]. This approach is in line with the recently published randomised trial Uterus 11 confirming the significant rate of surgical upstaging (33%) in FIGO stages T2b-4 of cervical cancer compared to clinical/radiological staging only [102]. Although statistical significance could not be reached, surgical staging in locally advanced cervical cancer led to superior disease-free survival (DFS) and overall survival (OS) compared to radiological staging only, with a significant benefit for patients with FIGO stage IIB [103]. The postoperative complication rate was low, and there was no significant delay in final treatment [104]. The FRANCOGYN study group provided evidence of the therapeutic value of surgical paraaortic staging in locally advanced cervical cancer with no involvement of paraaortic lymph nodes on pre-operative imaging examination [105]. In multivariate model analysis, surgical staging remained an independent prognostic factor for DFS (OR 0.64, CI 95% 0.46–0.89, *p* = 0.008) and OS (OR 0.43, CI 95% 0.27–0.68, *p* < 0.001). Being aware of not only the false negative but also the false positive results using any imaging modality, any equivocal suspicion of extrauterine disease spread, which would change patient management, should be verified by a biopsy to avoid inappropriate treatment. In such cases, core-needle (tru-cut) biopsy is the preferred option to a fine-needle aspiration biopsy. Core-needle biopsy uses a thicker gauge needle and, therefore, retrieves representative biopsy samples and allows for histological diagnosis [9]. The biopsy is usually guided by ultrasound or, in less assessable sites, by CT [106,107]. The positive findings are confirmative of diagnosis, but negative findings should be taken with caution (cannot exclude inadequate sample) and should be assessed altogether with the results of imaging.

To sum up the nodal and distant staging, the joint ESGO-ESTRO-ESP cervical cancer guidelines recommend surgicopathological staging of pelvic lymph nodes in early-stage disease with negative lymph nodes on imaging. In locally advanced cervical cancer (T1b3 and higher (except T2a1) or in early-stage disease with suspicious lymph nodes on primary work-up imaging (DWI-MRI or ultrasound), whole-body PET-CT or a chest/abdomen/pelvis contrast-enhanced CT for detecting nodal and distant disease are recommended. PET-CT is recommended before definitive chemoradiation with curative intent [9]. The reason for preferring PET-CT before CT in the latter group is the superiority of PET-CT to assess nodal or distant metastatic spread by its detection of increased FDG avidity in metastatic lesions [66,92,108]. It can also be used to evaluate the treatment effect and differentiate between persistent tumour and postradiotherapy changes. In addition to PET-CT or thoracic and abdominal CT in locally advanced cervical cancer, surgicopathological staging consisting of paraaortic lymph node dissection may be considered in women with negative paraaortic lymph nodes on imaging. Such an approach should reduce the risk of unrecognised occult small-volume metastases [9]. Moreover, to avoid the risk of inappropriate treatment due to false-positive imaging findings, any equivocal extrauterine spread findings should be verified by an image-guided tru-cut (core needle) biopsy if possible.

## 5. Conclusions

Accurate cervical cancer staging is essential for individualised treatment planning in order to achieve the best treatment results, and imaging findings are formally incorporated in stage assignment. Ultrasound or MRI is recommended in the primary diagnostic workup for local staging of cervical cancer. Ultrasound is cost-efficient, widely affordable, and a non-invasive imaging method with comparable diagnostic performance to that of MRI for preoperative tumour detection and assessment of local tumour extension (size of tumour, stromal and parametrial invasion, bladder, and rectal infiltration). Both imaging methods are also highly specific in identifying metastatic pelvic lymph nodes, with similar diagnostic performance, but larger prospective studies with head-to-head comparisons are necessary to verify this. In the early stages (up to T2a1, excluding T1b3) with negative pelvic lymph nodes on ultrasound or MRI, surgicopathological staging of pelvic lymph nodes is the method of choice due to the poor sensitivity of any imaging modality to detect small-volume metastases. In locally advanced stages (T1b3 and higher, excluding T2a1) or early stages with suspicious lymph nodes on ultrasound or MRI, PET-CT or whole-body CT should be part of the routine preoperative workup to detect nodal and distant tumour spread. Paraaortic surgicopathological staging in patients with negative paraaortic lymph nodes on primary imaging may be considered to exclude small-volume lymph node metastases otherwise undetectable.

## Figures and Tables

**Figure 1 cancers-16-00775-f001:**
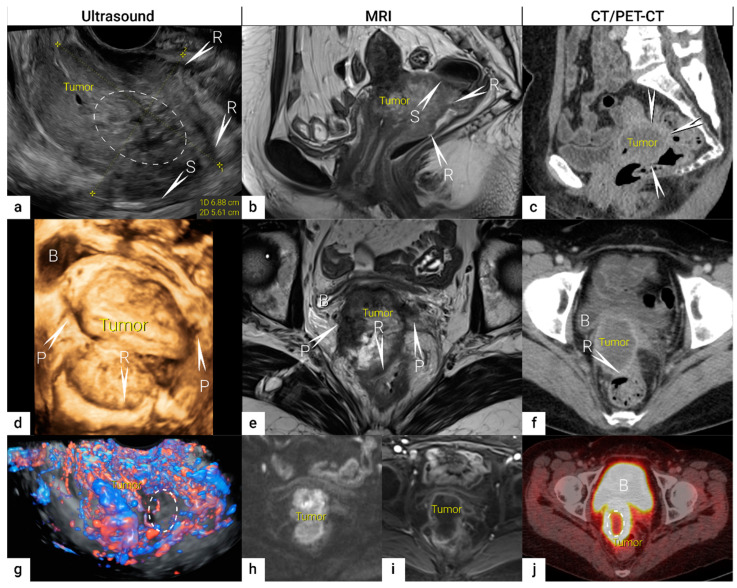
Bulky cervical squamous cell carcinoma in a 48-year-old woman with FIGO stage IVA (T4 N1 M0).Upper row: a 7 cm large (maximum diameter), mostly necrotic tumour depicted in the sagittal plane by (**a**) transrectal US (hypoechogenic tumour marked with dotted line having central irregular necrotic cystic areas (ellipse); (**b**) T2-weighted MRI depicting a hyperintense cervical lesion and contrast-enhanced (CE) CT (**c**) depicting a hypodense tumour. The infiltration of the rectal wall (R) and sigmoid colon (S) are marked with arrows and single letters of corresponding adjacent organ infiltration. Middle row: the same tumour depicted in the transverse plane by transrectal three-dimensional ultrasound (**d**); T2-weighted MRI (**e**) and CE CT (**f**); exhibits infiltration of lateral parametria (P) and rectum (R) marked with arrows and single letters of corresponding adjacent organ infiltration. Lower row: three-dimensional colour Doppler US (**g**) depicts high perfusion within the same tumour except in the central necrotic part (ellipse). Paraaxial DWI (**h**) (high b-value image: b = 1000) depicts restricted diffusion within the tumour and contrast-enhanced MRI (DCE-MRI) (**i**) enhancement rim in the periphery of the lesion with central necrosis. Axial FDG- PET-CT (**j**) depicts FDG-avid tumour except for the central necrotic part (ellipse). B, bladder; FDG, fluorodeoxyglucose; FIGO, the International Federation of Gynaecology and Obstetrics; CT, computed tomography; MRI, magnetic resonance imaging; P, lateral parametria; PET-CT, positron emission tomography fused with computed tomography; R, rectum; S, sigmoid colon.

**Figure 2 cancers-16-00775-f002:**
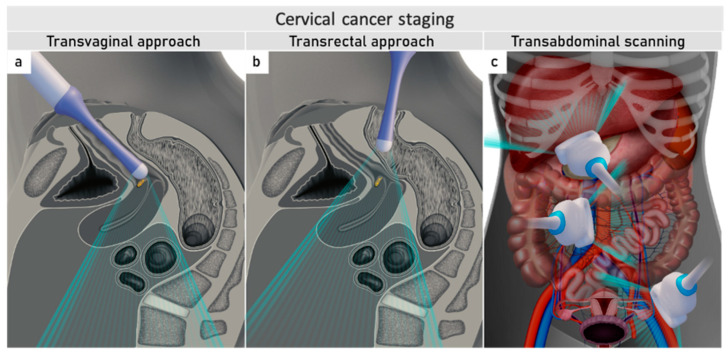
Ultrasound for cervical cancer staging.Transvaginally inserted probe (**a**). Transrectally inserted probe (**b**). Transabdominal scanning (**c**).

**Figure 3 cancers-16-00775-f003:**
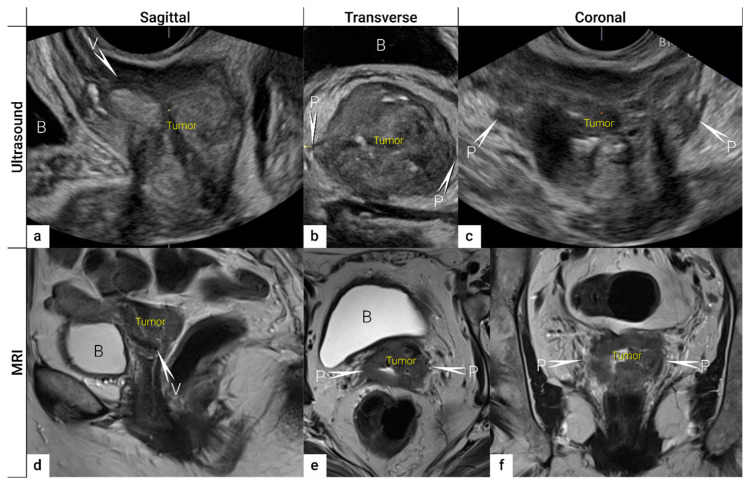
Three imaging planes of three-dimensional (3D) ultrasound and MRI. Cervical squamous cell carcinoma in a 57-year-old woman with FIGO stage IIIC1 (T2b N1 M0). Sagittal (**a**), transverse (**b**), and coronal plane (**c**) on 3D ultrasound depict a large tumour of 5 cm (max diameter) as a hypoechogenic mass. T2-weighted MRI images in sagittal (**d**), transverse (**e**), and coronal (**f**) planes visualise the tumour as a slightly hyperintense cervical mass. Both imaging methods showed in the sagittal view an infiltration of the ventral vaginal wall (V) and in the transverse and coronal plane a bilateral invasion of lateral parametria (P) marked with arrows and single letters of corresponding adjacent soft tissue infiltration. B, bladder; MRI, magnetic resonance imaging; V, vagina.

**Figure 4 cancers-16-00775-f004:**
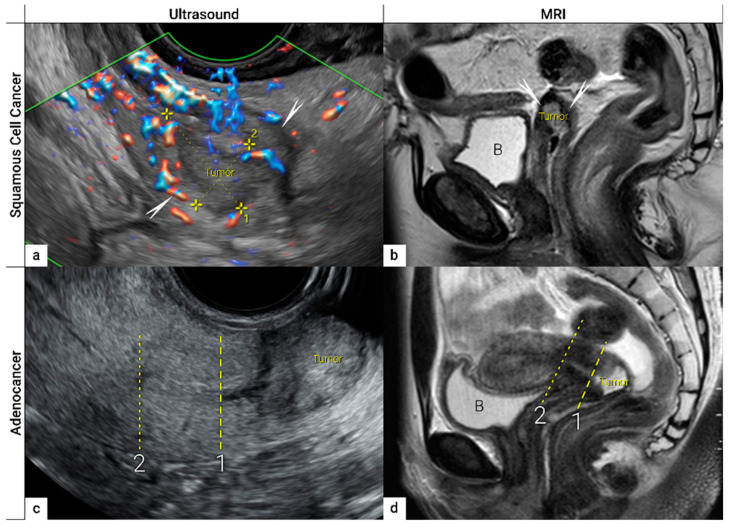
Imaging of cervical squamous cell carcinoma and adenocarcinoma. Upper row: cervical squamous cell cancer in the remaining cervix after previous supravaginal hysterectomy (due to uterine prolapse) in a 59-year-old woman with clinical FIGO stage IB1 (T1b1 N0 M0), depicted in the sagittal plane by transrectal ultrasound (**a**) as a 2 cm large (maximum diameter) hypoechoic tumour marked with callipers and surrounded by a rim of a non-involved hyperechoic stroma (arrows), sagittal T2-weighted MRI (**b**) depicting a hyperintense cervical lesion surrounded by hypointense normal cervical stroma (arrows) in the same patient. Lower row: adenocarcinoma stage T1b1 N0 M0 located on the exocervix in a 28-year-old woman before fertility-sparing surgery and depicted in the sagittal plane by transrectal ultrasound (**c**) as a 3 cm large hyperechogenic tumour and by T2 weighted MRI (**d**) as a hyperintense tumour. 1- cranial tumour margin; 2- internal os; distance between the dotted and dashed line corresponds to the cranial tumour-free margin. B, urinary bladder.

**Figure 5 cancers-16-00775-f005:**
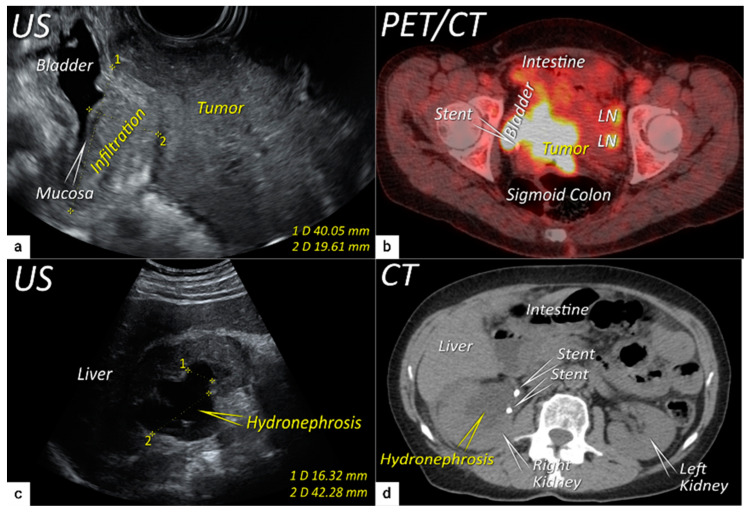
Infiltration of the right ventral parametria and hydronephrosis. The sagittal plane of the transrectal ultrasound (US) (**a**) demonstrating infiltration of the right ventral parametrium and the bladder wall (except for mucosa marked with an arrow) by a large squamous cell cervical carcinoma of 94 mm in the largest diameter in a 65-year-old woman with T3bN1M0/FIGO stage IIIC1. The transverse plane of fluorodeoxyglucose (FDG) -positron emission tomography-computed tomography (PET-CT) (**b**) in the same patient demonstrating infiltration of the bladder and sigmoid colon wall and metastatic lymph nodes in the left obturator fossa (LN). The sagittal plane of a transabdominal ultrasound scan in the same patient (**c**) demonstrating right kidney hydronephrosis (second grade) due to locally advanced cervical carcinoma with partial obstruction of the right ureter causing dilated renal calyx (1) and pelvis (2). The transverse plane of non-contrast CT (part of the FDG PET-CT examination) depicts second-grade hydronephrosis in the right kidney after placement of a ureteral stent in the same patient (**d**). CT, computed tomography; FDG PET-CT, fluorodeoxyglucose positron emission tomography fused with computed tomography; US, ultrasound.

**Figure 6 cancers-16-00775-f006:**
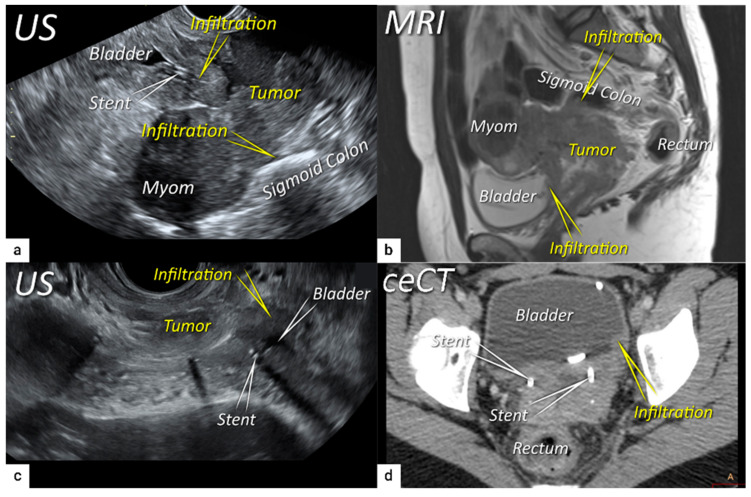
Infiltration of bladder and rectosigmoid colon wall. The sagittal plane of a transrectal ultrasound (US) (**a**) and T2-weighted magnetic resonance imaging (MRI) (**b**) depicting a squamous cell cervical carcinoma with necrotic components in a 38-year-old woman (T4N1M0/FIGO stage IVA) having tumour infiltrating the bladder and sigmoid colon wall including mucosa. Transverse plane of transrectal ultrasound (**c**) and axial contrast-enhanced computed tomography (**d**) showing deep bladder invasion in the same patient treated with bilateral ureteral stents. The deepest infiltration was seen on the left side of the bladder.

**Figure 7 cancers-16-00775-f007:**
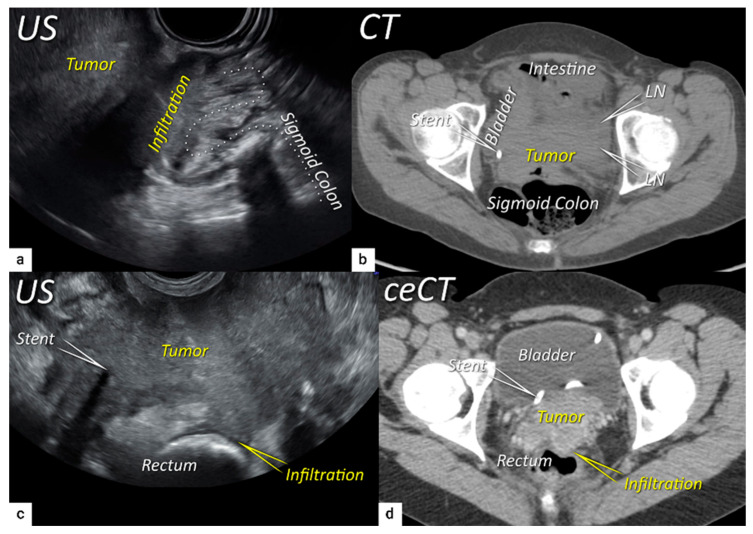
Infiltration of the rectosigmoid wall. Upper row: Sagittal transrectal ultrasound depicting a large squamous cell cervical carcinoma with a maximum diameter of 94 mm (T3bN1M0/FIGO stage IIIC1) in a 65-year-old woman (same patient as Figure 5), infiltrating the rectosigmoid colon with retraction of loops towards the tumour (**a**). Axial non-contrast computed tomography (CT) (**b**) shows a tumour infiltrating the muscle wall of the sigmoid colon and bladder and the pelvic side wall bilaterally with bulky lymph nodes in the left obturator fossa (LN). The right ureter with a stent is embedded within the right lateral pelvic side wall invasion. Lower row: A 38-year-old patient with maximum diameter 48 mm (T4N1M0/FIGO stage IVA) (same patient as in Figure 6) with squamous cell carcinoma depicted in the transverse plane by transrectal ultrasound (**c**) showing infiltration of the muscle layer of the rectum from the left uterosacral ligament and bilateral pelvic side wall infiltration with a visible right ureteral stent within the pelvic side wall infiltration. The transverse plane of contrast-enhanced computed tomography (CT) (**d**) demonstrates rectal wall infiltration and ureteral stent laterally. US, ultrasound; MRI, magnetic resonance imaging; ceCT, contrast-enhanced computed tomography.

**Figure 8 cancers-16-00775-f008:**
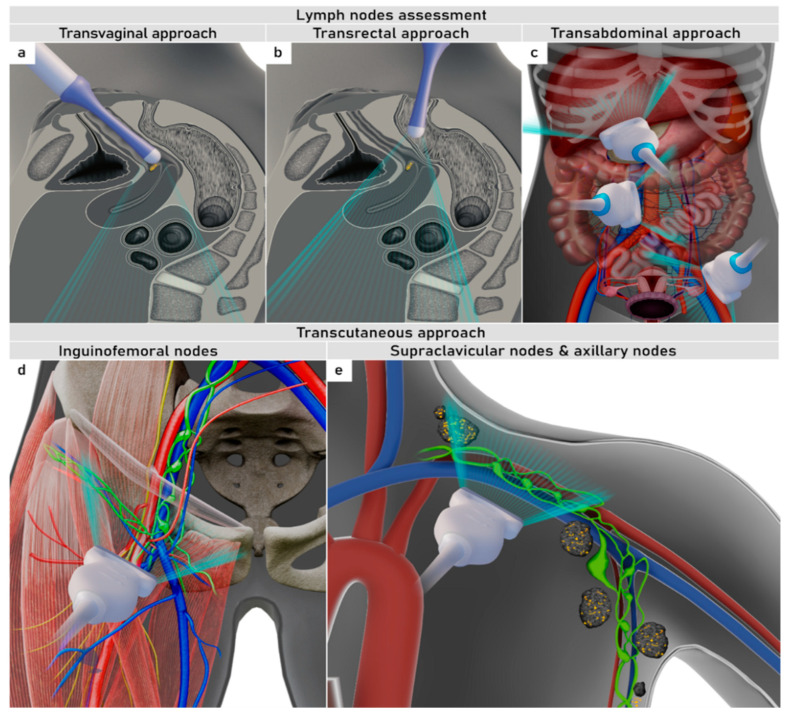
Schematic diagram showing ultrasound approach to infra- and supradiaphragmatic lymph nodes. TAS, transabdominal scan; TRS, transrectal scan; TVS, transvaginal scan.

**Figure 9 cancers-16-00775-f009:**
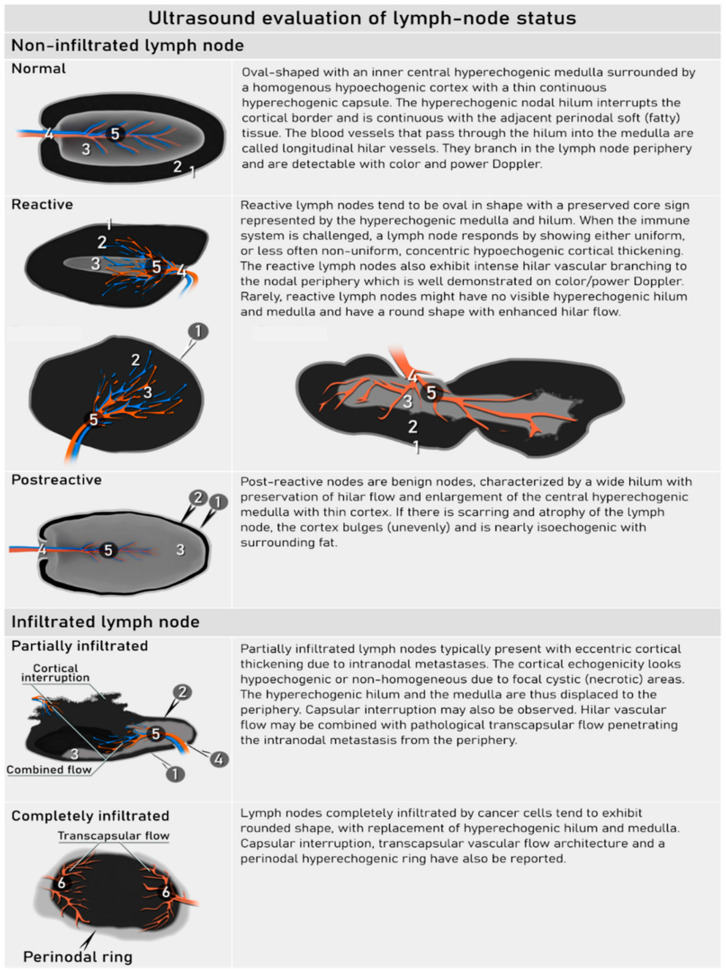
Ultrasound features of normal, reactive and infiltrated lymph nodes.

**Figure 10 cancers-16-00775-f010:**
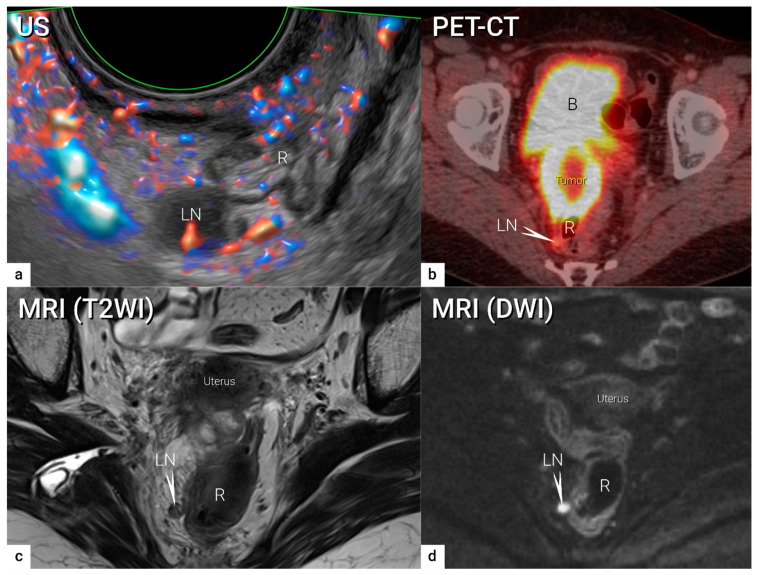
Visceral metastatic lymph node in the right uterosacral ligament (grade 4 parametrial infiltration). A visceral metastatic lymph node of size 10 mm pararectally in the right uterosacral ligament in a 48-year-old patient with bulky cervical carcinoma of maximum diameter 69 mm, infiltrating the rectum and sigmoid colon (same as in Figure 1) is depicted in the transverse plane by transrectal Doppler ultrasound (**a**) showing rounded metastatic hypoechogenic lymph node (LN5) with mixed perfusion (longitudinal hilar vessels and peripheral flow), axial PET-CT with rounded slightly FDG-avid lymph node (arrow) (**b**), paraaxial T2-weighted MRI (T2WI) (**c**) depicting slightly hyperintense round lymph node (arrow) and paraxial high b-value diffusion-weighted MRI (DWI) (**d**) depicting restricted diffusion in the same lymph node (arrow) (primary tumour is not visible in this plane). B, bladder; DWI, diffusion-weighted imaging; FDG, fluorodeoxyglucose; LN, lymph node; MRI, magnetic resonance imaging; PET-CT, positron emission tomography fused with computed tomography; R, rectum; T2WI, T2-weighted imaging; US, ultrasound.

**Figure 11 cancers-16-00775-f011:**
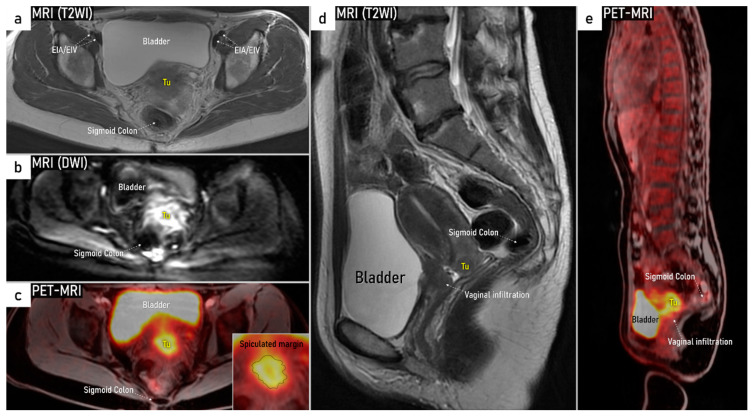
PET-MRI imaging of uterine cervix carcinoma. A 40-year-old woman (T2bN0M0/FIGO stage IIB) having a squamous cell cervical carcinoma with lateral parametrial infiltration bilaterally and infiltration of the upper third of the vagina. The rectum and urinary bladder are intact. No pathologic lymph nodes are seen in the pelvis or at other localisation. The axial T2W imaging (MRI) (**a**), diffusion-weighted imaging (DWI; high b-value image) (**b**) and fluorodeoxyglucose (FDG)-positron emission tomography (PET)-MRI (**c**) depicting cervical carcinoma (tumour) with spiculated margins bilaterally infiltrating the parametria (arrows). The tumour exhibits diffusion restriction with hyperintensity on the high b-value image (DWI) and high FDG-avidity depicted by PET-MRI. The sagittal T2W image (**d**) and PET-MRI (**e**) depict the tumour as hypermetabolic (PET-CT) and hyperintense (T2W image) and affecting both the endo- and exocervix (tumour) and infiltrating the upper third of the vagina (arrow).

**Table 1 cancers-16-00775-t001:** Revised FIGO 2018 and TNM staging of cervical cancer and the role of imaging [11,15].

TNM Category ^a^	FIGO Stage ^a^	Criteria	Imaging Findings
T (TUMOUR) ^b^		T CRITERIA	Tumour delineation by imaging
TX		Primary tumour cannot be assessed	Non-diagnostic imaging examination US: Acoustic limitations in depicting the tumour due to e.g., abnormal position of uterus, severe pelvic infection with pyometra/actinomycosis/tuberculosis MRI: Severe imaging artefacts due to e.g., patient movement or hip implants
T0		No evidence of primary tumour	No visible tumour depicted by imaging US: Regular contours of exocervix and endocervical canal, homogeneous echogenicity of cervical stroma, regular arrangement of cervical vessels and pericervical fascia. MRI: The four cervical zones are depicted on T2W images with high signal intensity endocervical canal, intermediate signal intensity plicae palmatae, low signal intensity fibrous stroma, and intermediate signal intensity outer stromal ring.
T1		Carcinoma is strictly confined to the cervix (extension to the corpus should be disregarded)	Tumour confined to the cervix (not necessarily visible in T1a); assessment of maximum tumour diameter is critical to substage T1b1-T1b3. The measurement of lateral +/− cranial tumour-free distance is optional
T1a ^c^	IA	Carcinoma with maximum depth ≤5 mm	Limited resolution for US or MRI to detect T1a cancers (invasive cancer diagnosed by microscopy)
T1a1	IA1	Stromal invasion ≤3 mm (diagnosed in biopsy)	
T1a2	IA2	Stromal invasion >3 mm and ≤5 mm (diagnosed in biopsy)	
	IB	Carcinoma with deepest stromal invasion >5 mm, limited to the cervix uteri with size measured by maximum tumour diameter.	US: Highly vascularised hypoechogenic (squamous-cell carcinoma) or iso-/hyperechogenic lesion (adenocarcinoma) with intact hyperechogenic pericervical fascia and positive sliding sign between tumour and bladder/rectum. MRI: Tumour has intermediate to high signal on T2W images. Dynamic CE-T1W images depicts tumour as hyperintense in the arterial phase and iso- or hypointense in the venous phase. Tumours characteristically exhibit restricted diffusion on DWI (hyperintensity on high b-value images and low intensity on the ADC maps). Tumour does not disrupt the hypointense peripheral stromal ring (best seen on T2W images).
T1b1 ^d^	IB1 ^d^	Carcinoma with >5 mm depth of stromal invasion and ≤2 cm in greatest dimension.	
	IB2	Carcinoma >2 cm and ≤4 cm in greatest dimension.	
	IB3	Carcinoma >4 cm in greatest dimension.	
T2	II	Carcinoma invades beyond the uterus but does not extend to the lower one-third of the vagina or to the pelvic wall	Tumour extends beyond the cervix with infiltration of upper two-third of vagina or pericervical fascia by US or MRI. Confident diagnosis of parametrial invasion is made in the presence of the spiculated tumour-parametrial interface, soft tissue mass in parametrium, encasement of periuterine vessels, and ureter.
T2a	IIA	Involvement limited to the upper two-thirds of the vagina without parametrial invasion	US: Highly vascularised tumour infiltrates upper two third of hypoechogenic vaginal wall, maximum diameter of primary tumour is critical to substageT2a1/T2a2. MRI: Involvement of the upper two-third of the vagina is seen as segmental loss of the T2-hypointensity of the vaginal wall.
T2a1	IIA1	Carcinoma ≤4 cm in greatest dimension	
T2a2	IIA2	Carcinoma >4 cm in greatest dimension	
T2b	IIB	Parametrial tumour invasion but no pelvic side wall extension	US: Tumour infiltrates the hyperechogenic pericervical fascia, negative sliding sign, presence of hypoechogenic tumour projections into hyperechogenic parametria. MRI: Tumour disrupts the hypointense peripheral stroma and extends into the parametrium +/− abutting parametrial vessels on T2W images.
T3 ^e^	III ^e^	Carcinoma involves the lower third of the vagina and/or extends to the pelvic side wall and/or causes hydronephrosis or non-functioning kidney	Tumour infiltration of the lower third of vagina or lateral pelvic side wall by US or MRI. Pelvic side wall infiltration is considered when the tumour causes hydroureter, infiltrates the obturator internus, piriformis, and levator ani muscles, encases the iliac vessels, or invades the pelvic bones on US or MRI.
T3a	IIIA	Carcinoma involves the lower third of the vagina, with no extension to the pelvic wall.	US: Highly vascularised, irregular tumour infiltration of the lower third of vagina MRI: Involvement of the lower third of the vagina is suggested by disruption of the normal low-signal-intensity wall on T2- weighted images
T3b	IIIB	Tumour extension to the pelvic side wall and/or hydronephrosis or non-functioning kidney (unless known to be due to other cause).	US: Hypoechogenic tumour projections up to pelvic side wall +/− infiltration of iliac vessels, ureters, muscles, presence of hydronephrosis MRI: Hyperintense infiltration up to the pelvic side wall, loss of normal parametrial signal intensity and increased signal intensity in pelvic musculature due to tumour invasion seen on T2W-images.
T4 ^f^	IVA ^f^	Tumour invasion into the mucosa of the bladder or rectum (biopsy-proven) or into adjacent organs.	Tumour invasion into the mucosa of the bladder or rectum on imaging, confirmed by biopsy. US: Negative sliding sign, hypoechogenic tumour infiltration of bladder/rectal wall up to echogenic mucosa with polypoid tumour seen intraluminally. MRI: Focal or diffuse disruption of the normal T2-low signal intensity wall of the bladder/rectum, irregular or nodular wall, sometimes including an intraluminal tumour mass. Bulous edema sign, which is hyperintense thickening of the bladder mucosa on T2W images, is only an indirect sign of invasion and should not be regarded as T4 unless confirmed mucosal infiltration at cystoscopy. Infiltration of the posterior bladder wall without mucosal infiltration should not be regarded as T4a.
N (NODE) CATEGORY ^g^			
Nx		Regional lymph nodes cannot be assessed	US: Poor acoustic conditions in the pelvis or abdomen due to tumour infiltrating pelvic side wall or patient obesity. MRI/CT/PET-CT: Non-diagnostic image quality due to severe artefacts from e.g., hip implants or patient movement.
N0		No regional lymph node metastasis	Demonstration of paracervical, parailiac and paraaortic tissue without suspicious lymph node/-s. US: Oval-shaped nodes, the inner central hyperechogenic medulla and hilum (nodal-core sign) is surrounded by a homogeneous hypoechogenic cortex and thin continuous hyperechogenic capsule. Hilar longitudinal vessels may be visualised. In reactive lymph nodes due to the activation of the immune response, uniform concentric cortical thickening of the hypoechogenic cortex and intesified normal vascular tree from the central hilar region may be found. MRI/CT/PET-CT: No suspicion of malignant lymph nodes due to normal sized lymph nodes without irregular contours/signal or signs of restricted diffusion (MRI/DWI) or pathologic FDG-avidity (PET-CT)
N0(i+) ^h^		Isolated tumour cells in regional lymph node(s) ≤0.2 mm or single cells or clusters of cells ≤200 cells in a single lymph node cross-section	
N1	IIIC1	Regional lymph node metastasis to pelvic lymph nodes only	US: hypoechogenic rounded lymph node without preservation of typical architecture (loss of the nodal core-sign), inhomogeneous echogenicity due to cystic necrosis and calcifications, capsular interruption, grouping of metastatic lymph nodes and others. Hilar flow may still be preserved in a partial nodal involvement with or without transcapsular vascularisation (vessels penetrating the cortex from outside), the latter are usually found in an advanced stage of infiltration. MRI: lymph nodes with maximum transverse diameter >10 mm; capsule irregularity, rounded (as opposed to oval) shape, inhomogeneous signal with signs of necrosis on T2W images (MRI), restricted diffusion (DWI) or increased FDG-avidity (PET-CT)
N1mi ^i^		Regional lymph node metastasis (>0.2 mm but ≤2 mm in greatest dimension) to pelvic nodes	Limited resolution for imaging
N1a		Regional lymph node metastasis (>2 mm in greatest dimension) to pelvic lymph nodes	Paracervical and parametrial nodes are first to be involved, followed by spread to external iliac nodes by lateral route, internal iliac nodes by hypogastric route, and lateral sacral and sacral promontory nodes by presacral route. All nodal groups drain to the common iliac nodes.
N2	IIIC2	Regional lymph node metastasis to para-aortic lymph nodes, with or without positive pelvic lymph nodes.	
N2mi ^i^		Regional lymph node metastasis (>0.2 mm but ≤2.0 in greatest dimension) to para-aortic lymph nodes, with or without positive pelvic lymph nodes	Limited resolution for imaging
N2a		Regional lymph node metastasis (>2.0 in greatest dimension) to para-aortic lymph nodes, with or without positive pelvic lymph nodes	Pelvic lymph nodes drain to the paraaortic nodes.
M (METASTASIS) CATEGORY			
M0		No distant metastasis	US: Systematic abdominal scanning, evaluation of groins and supraclavicular lymph nodes without abnormal findings MRI/CT/PET-CT: No signs of metastatic lesions in the abdomen (MRI) or in the thorax/trunk/groins/head (whole-body PET-CT)
cM1 ^j^	IVB	Distant metastasis (clinical category)	US: non-regional lymph node infiltration (supraclavicular, inguinal, and other regions), hypoechogenic or target (hypoechogenic rim and echogenic center) liver lesion/-s, hypoechogenic infiltration of suprarenal glands, hypoechogenic peritoneal carcinomatosis and others. MRI, CT, PET-CT: distant spread of tumour to the liver, lung, bones, peritoneum, and soft tissue, and rarely adrenals, spleen, kidneys, pancreas, and gastrointestinal tract and others.
pM1 ^j^	IVB	Distant metastasis (pathologic category)	

(a) All imaging modalities and pathology can be used, when available, to supplement clinical findings with respect to tumour size and extent, in all stages. Pathological findings supersede imaging and clinical findings. (b) Involvement of lymphovascular spaces should not change the staging but may affect the treatment plan. (c) The diagnosis of T1a1,2 is made by microscopic examination of a surgical specimen, which includes the entire lesion. The depth of invasion should not be greater than 3 or 5 mm, respectively, from the base of the epithelium. For T1a1,2 the horizontal dimension is no longer considered in defining the upper boundary of a T1a carcinoma [11]. The margins of a cone specimen should be reported to be negative for disease to do the final pathological stage. If the margins of the cone biopsy are positive for invasive cancer, the patient is assigned to T1b1. Exceptions are allowed, for example, for large exophytic tumours >2 cm in the largest dimension, which should be staged based on the largest dimension even if they are superficially invasive (≤5 mm) (https://www.iccr-cancer.org/wp-content/uploads/2022/02/ICCR-Cervix-4th-edn-v4-bookmark.pdf, Accessed on 12 January 2024). In some situations (e.g., in ulcerated tumours) it is not possible to measure the depth of invasion. In such cases, the tumour thickness may be measured, and this should be clearly stated in the pathology report, together with the reasons why the thickness and not the depth of invasion is given. In such cases, the pathologist and clinician should correlate tumour thickness with depth of invasion for staging and management purposes (https://www.iccr-cancer.org/wp-content/uploads/2022/02/ICCR-Cervix-4th-edn-v4-bookmark.pdf, Accessed on 12 January 2024). The expert panel recommends that for all morphological subtypes, the term ‘microinvasive carcinoma’ should be avoided, and instead, the use of specific TNM and FIGO stages is recommended [9]. (d) A new primary tumour size cut-off value of 2 cm enables evaluating potential candidates for fertility-sparing treatment. For this purpose, craniocaudal cervical length and tumour-to-internal cervical os distance are also measured. (e) The pelvic wall is defined as the muscle, fascia, neurovascular structures, and skeletal portions of the bony pelvis. (f) Bullous oedema does not permit a case to be assigned to stage IVA. (g) Adding notation of r (radiology) and p (pathology) to indicate the findings that are used to allocate the case to Stage IIIC (e.g., IIICp, IIICr). The type of pathology technique used should also be documented. The suffix (f) is added to the N category when metastasis is identified by fine-needle aspiration or core biopsy. The suffix (sn) is added to the N category when metastasis is identified only by sentinel lymph node biopsy. When in doubt, the lower staging should be assigned. (h) Isolated tumour cells do not change the stage, but their presence should be recorded. (i) Micrometastases are included in Stage IIIC. Isolated tumour cells do not change the stage, but their presence should be recorded. j Includes metastasis to inguinal, mediastinal, supraclavicular, and other lymph nodes regions beyond the abdomen, intraperitoneal disease, lung, liver, or bone, excludes metastasis to pelvic or para-aortic lymph nodes or vagina. ADC, apparent diffusion coefficient; CE-T1W, contrast-enhanced T1 weighted images; CT, computed tomography; DWI, diffusion-weighted imaging; FIGO, International Federation of Gynecology and Obstetrics staging of cancer of the cervix; FDG, fluorodeoxyglucose; MRI, magnetic resonance imaging; PET-CT, positron emission tomography-CT; T2W, T2 weighted images; US, ultrasound. Underlined sentences indicate key criteria in respect of imaging for each T-category.

**Table 2 cancers-16-00775-t002:** Comparison of Different Imaging Methods for Application in Gynecologic Oncology.

Parameters	Expert Ultrasound	MRI	CECT	FDG-PET-CT
Cost	1×	4×	2×	6×
Availability	Specialised centers	Most hospitals	Most hospitals	Specialised centers
Range of examination	Abdomen and pelvis, peripheral lymph nodes	Whole body	Whole body	Whole body
Examination duration (minutes)	15–30 ^#^	30–45 (pelvic) 60 (whole-body) ~15 (reading time)	5 ~10 (reading time)	30 ~20 (reading time)
Dynamic evaluation *	Yes	No	No	No
Preparation before imaging	None	Antiperistaltic agents	None	4 h fasting and 1 h rest prior to scanning
Contrast agent	None **	Gadolinium-based	Iodine-based	FDG-radiotracer and iodine-based
Radiation exposure	None	None	10–20 mSv ^ß^	20–25 mSv ^ß^
Limitations for application and factors impacting diagnostic quality	No contraindications. Limited depiction of abdominal deeper structures when overlying bowel gas/air	Contraindication if severe claustrophobia, and for some metal- or cochlear implants/cardiac pacemakers. Image artefacts from implants.	Contraindication for iodine-based contrast agent: - renal insufficiency ^&^ - hyperthyroidism - severe iodine allergy Image artefacts from implants.	Contraindication for iodine-based contrast agent: - renal insufficiency ^&^ - hyperthyroidism - severe iodine allergy Image artefacts from implants.
Dependence of expertise	Yes	Yes	Yes	Yes

^#^ Examination time depends on stage of disease (in advanced stage there is more complex checklist including scalene lymph nodes); * Ultrasound imaging provides information about site-specific tenderness and visualises how pelvic structures move in relation to each other (positive sliding sign or negative sliding sign in adhesions); ** Contrast-enhanced ultrasound is not recommended in gynecology following EFSUMB guidelines (update 2017) [57]; ^ß^ Radiation exposure of 10–20 mSv from CECT (thorax, abdomen, pelvis) equals ~3–7 years of average background radiation and radiation exposure of 20–25 mSV from FDG-PET-CT (whole body) equals ~7–8 years of average background radiation (https://www.acr.org/-/media/ACR/Files/Radiology-Safety/Radiation-Safety/Dose-Reference-Card.pdf, Accessed on 12 January 2024). ^&^ In patients with renal insufficiency gadolinium contrast media must be used with caution; CECT, contrast-enhanced computed tomography; EFSUMB, European Federation of Societies for Ultrasound in Medicine and Biology; FDG-PET-CT, 18F-fluorodeoxyglucose positron emission tomography combined with CT; MRI, magnetic resonance imaging; mSv, millisievert.

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
