# Peer review of "The Role of Imaging in Cervical Cancer Staging: ESGO/ESTRO/ESP Guidelines (Update 2023)"

_cancers, 2024, doi:10.3390/cancers16040775_

Round 1
Reviewer 1 Report
Comments and Suggestions for Authors
The article is a review on the role of imaging in cervical cancer staging with a focused interest on ultrasound. It is a comprehensive and thorough review that is of interest to a broad audience working on cervical cancer. However, the "focus" on ultrasound creates a bit of bias in the article, and I would ask the authors to include a bit more on the weaknesses of ultrasound.
For example:
1) Having an expert on UL available (time consuming for MD compared to looking at MR images) 2) Reproducability of UL compared to MR/CT/PET 3) (MR/CT/PET images easily available for multidisciplinary team meetings) Ultrasound not. 4) Ultrasound as a tool in radiation treatment planning highly debated. 5) problems with air cavities for UL
The authors also summarises many of the chapters with comparing MR and UL, however the comparison seems biased. The references used are very old 2004-2012. In the rapid development of medical imaging this is outdated information. If there are no new articles on this subject, the authors should highlite the need for more updated information. One of the most used references used for comparing MR and UL are from 2006 [reference 4]! Hence, MR T2-weighted images are compared to UL. Most senters use DWI and DCE often in combination with FDG-PET /CT. All these techniques are complementary and provide independent information on the disease burden. A recent reference to this would be: https://pubmed.ncbi.nlm.nih.gov/27528120/
In summary, the review is comprehensive, well written and interesting, but the authors could provide a bit newer references or highlite the need for new clinical studies providing a more updated information.
Author Response
Dear reviewer,
We appreciated and considered your critical suggestions; we thank you for sharing them.
1.Regarding the reported bias on ultrasound, we originally invited an MRI expert, professor Ingfrid S. Haldorsen from University of Bergen to co-author this review trying to make it more balanced. In chapter 3 “Local (pelvic) workup for different stages”, lines 265-269, we had already mentioned the reported limitations of ultrasound. However, we understand the criticism that has been raised, which is why we have tried to accommodate the request by adding a few sentences at the end of the paragraph. You will find them at lines: 325-338
2. Regarding the outdated references, there are very little studies comparing different modalities in cervical cancer in one cohort (head to head comparison) prospectively. It explains why such comparative studies were mentioned. Regarding T2-MRI /DCE and ultrasound, the last comparative study by Stukan M. et al. from 2021 is mentioned (Accuracy of Ultrasonography and Magnetic Resonance Imaging for Preoperative Staging of Cervical Cancer-Analysis of Patients from the Prospective Study on Total Mesometrial Resection - PubMed (nih.gov)).
3.Thank you for your remark on the use of functional MRI in brachytherapy planning, we have tried to expand them with new references and a small paragraph, you will find it at lines: 251-255 Our opinion remains that there is a need for prospective studies and more robust data, we have pointed out this shortcoming: “Nevertheless, validation through robust prospective data, prior to extensive adoption of DWI and DCE-MRI in cervical cancer, is essential and particular attention must be paid to the use of uniform protocols, standardized nomenclature and correlation of imaging findings with histopathology “
4.Regarding future research there is there is a note in part 4 Nodal work-up ‘All these limitations lead to heterogeneous data and, therefore, difficulties drawing meaningful conclusions. For this reason, a multicentre prospective study comparing ultrasound, DWI-MRI and FDG PET-CT in nodal staging has been initiated and the results are expected in 2025 (CANNES study).‘
Reviewer 2 Report
Comments and Suggestions for Authors
This submission represents a comprehensive review of current imaging modalities for cervical cancer staging. This review is very well composed and up to date.
Of particular interest is that the authors point to the several virtues of ultrasound in an evidence-based manner. Hence, not only is ultrasound a cost-effective approach, but as illustrated it’s capable of yielding practical, robust & occasionally superior data over other imaging modalities.
Figures are particularly instructional.
Lines 53-54: “The multicentric clinical trial” could be changed to “Twenty years ago, the multicentric clinical trial”.
Overall a particularly well structured submission which might prove very helpful for physicians like a) Gynecologists undergoing training in the subspecialty of Gynecologic Oncology, and b) General Gynecologists with a special interest in the field of Gynecologic Oncology.
Author Response
Dear reviewer,
We are grateful to hear that our work has been appreciated and deemed interesting.
We have added the few suggested words in the sentence you have pointed out. (Lines 53-54)
Thank you for your precious time
Reviewer 3 Report
Comments and Suggestions for Authors
Dear Authors,
manuscript is interesting and well written. My minor suggestion is, that heading of all tables should be always be at the top, not at the bottom as it is with figures. Moreover, text formatting in tables should be done.
The main question of this manuscript is to review the current diagnostic possibilities in cervical cancer. The latest methods used in the diagnosis of this neoplasm are described: CT, MRI and ultrasound are described and compared in a transparent way. The manuscript discusses these methods in relation to current guidelines and presents them in a clear way. References (109 position) are well selected regarding to the newest position in review field. Figures have appropriate quality, hovewer is in neccessary to correct text formating in tables, and also headline must be at the top of tables, this is a common rule in articles. In addition, text formatting of figure descriptions should be done (spacing of lines is too large).
Author Response
Dear reviewer,
We are grateful to hear that our work has been appreciated and deemed interesting.
We have made the changes you have pointed out: Tables 1 and 2 text was formatted, titles were moved above, all tables and figures descriptions have been reshaped according to the layout suggestions
Thank you for your precious time